# Rare loss of function variants in the hepatokine gene *INHBE* protect from abdominal obesity

Aimee M. Deaton [1✉], Aditi Dubey[1], Lucas D. Ward [1], Peter Dornbos[2,3,4], Jason Flannick [2,3,4], AMP-T2D-GENES Consortium*, Elaine Yee[1], Simina Ticau[1], Leila Noetzli[1], Margaret M. Parker[1], Rachel A. Hoffing[1], Carissa Willis[1], Mollie E. Plekan[1], Aaron M. Holleman[1], Gregory Hinkle[1], Kevin Fitzgerald[1], Akshay K. Vaishnaw[1] & Paul Nioi[1]

Identifying genetic variants associated with lower waist-to-hip ratio can reveal new therapeutic targets for abdominal obesity. We use exome sequences from 362,679 individuals to identify genes associated with waist-to-hip ratio adjusted for BMI (WHRadjBMI), a surrogate for abdominal fat that is causally linked to type 2 diabetes and coronary heart disease. Predicted loss of function (pLOF) variants in *INHBE* associate with lower WHRadjBMI and this association replicates in data from AMP-T2D-GENES. *INHBE* encodes a secreted protein, the hepatokine activin E. In vitro characterization of the most common *INHBE* pLOF variant in our study, indicates an in-frame deletion resulting in a 90% reduction in secreted protein levels. We detect associations with lower WHRadjBMI for variants in *ACVR1C*, encoding an activin receptor, further highlighting the involvement of activins in regulating fat distribution. These findings highlight activin E as a potential therapeutic target for abdominal obesity, a phenotype linked to cardiometabolic disease.

[1] Alnylam Pharmaceuticals, Cambridge, MA, USA. [2] Programs in Metabolism and Medical & Population Genetics, Broad Institute, Cambridge, MA, USA. [3] Division of Genetics and Genomics, Boston Children's Hospital, Boston, MA, USA. [4] Department of Pediatrics, Harvard Medical School, Boston, MA, USA. *A list of authors and their affiliations appears at the end of the paper. ✉email: adeaton@alnylam.com

The distribution of adipose tissue plays a role in metabolic health and cardiovascular disease risk that is independent of overall adiposity as assessed by body mass index (BMI). This relationship has been shown by both epidemiologic and genetic studies examining the impact of waist-to-hip ratio adjusted for BMI (WHRadjBMI), a surrogate for abdominal adiposity, on cardiometabolic disease and mortality[1–4]. Mendelian randomization (MR) studies have established a causal link between increased WHRadjBMI and risk of type 2 diabetes (T2D) and coronary heart disease (CHD) as well as glycemic traits, circulating lipids, and blood pressure[1,3].

The mechanisms influencing fat distribution in humans are not fully elucidated and few medical therapies specifically reduce visceral fat even though it is thought to confer considerable cardiometabolic risk[5,6]. Studying the genetic determinants of WHRadjBMI may offer insights into these mechanisms and identify potential drug targets[7,8]. Large GWAS have identified many common variants with small effects on WHRadjBMI and highlighted the impact of adipogenesis and insulin resistance on abdominal adiposity[9–12]. GWAS of imaging-derived measures of fat distribution such as visceral and abdominal subcutaneous adipose tissue volume have yielded additional insights[13]. A number of studies have connected rare genetic variation to fat distribution. An early sequencing study performed in women reported rare variants in IKBKB associating with waist-to-hip ratio (WHR)[14]. Rare variants in PDE3B and ACVR1C have been reported to associate with WHRadjBMI and related traits[9,10,13] and an exome array study of WHRadjBMI reported several novel rare variant associations[15]. A recent study using exome-sequences from 184,246 individuals uncovered novel associations with WHRadjBMI for loss of function in PLIN1, INSR and PLIN4[16]. Because the study of rare coding variation is a powerful method for identifying potential therapeutic targets[8,17–20], we used whole-exome sequencing data from 362,679 individuals to look for additional genes harboring variants with large effects on WHRadjBMI that may be candidates for therapeutic intervention.

## Results

**Exome-wide gene burden associations with waist-to-hip ratio adjusted for BMI.** We used whole exome-sequencing data from the UK Biobank (UKB)[21] to perform gene-based analysis of WHRadjBMI in 362,679 European ancestry individuals. We used three variant aggregation strategies: testing rare (MAF ≤ 1%) predicted loss of function (pLOF) variants, predicted damaging missense variants (missense), and the two combined (pLOF + missense) in up to 17,961 genes for association with WHRadjBMI. Twelve genes significantly associated with WHRadjBMI ($P \leq 1.05 \times 10^{-6}$; Methods) using at least one variant aggregation strategy, including PDE3B, ACVR1C, SLC5A3, and PLIN4 which have been reported to associate with fat distribution[9,10,16,22] and the Mendelian disease genes PLIN1, PYGM, and INSR which have been highlighted by other studies on WHRadjBMI[16,23–26]. For the remaining genes – COL5A3, ANKRD12, KEAP1, TRIM40, and INHBE – this study provides the first reported evidence linking rare coding variation to abdominal adiposity (Fig. 1, Supplementary Figs. 1 and 2, Table 1). Conditional analysis confirmed that rare variant associations were likely independent of each other (Supplementary Table 1) and independent of nearby common variant associations (Supplementary Table 2). TRIM40 was the only gene where study-level significance was lost when we conditioned on the top common-variant hit in the region ($P = 2.55 \times 10^{-6}$ compared to $P = 6.99 \times 10^{-7}$; Supplementary

Table 2). Given the proximity of TRIM40 to the HLA gene cluster and the partial dependence of the signal on common variants, we focused further analysis on the remaining 11 genes. Of the four genes not reported previously, only INHBE contained pLOF variants associated with lower WHRadjBMI (0.22 standard deviation (SD) decrease in WHRadjBMI; $P = 4.98 \times 10^{-8}$). INHBE, encoding inhibin subunit βE, is nearly exclusively expressed in the liver which lies in contrast to other WHRadjBMI-associated genes identified in this study which are enriched for adipose expression ($P = 1.44 \times 10^{-4}$; Supplementary Data 1).

We performed gene burden analysis of WHRadjBMI in the South Asian ($n = 7367$), East Asian, ($n = 1306$) and African ancestry ($n = 6129$) sub-populations of UKB for variant sets with sufficient power. No additional significant genes were identified when these sub-populations were meta-analyzed with the European ancestry population and no exome-wide significant associations were detected in the individual sub-populations. However, eight (67%) of the variant sets with power to test in the other sub-populations showed a consistent direction of effect and/or nominal significance ($P \leq 0.05$), including PLIN1 pLOF and ACVR1C missense variants (Supplementary Data 2).

A sex-specific analysis of WHRadjBMI identified associations for variants in GIGYF1 with increased WHRadjBMI ($P = 6.76 \times 10^{-7}$, Beta = 0.57 SD) and the lipodystrophy gene LIPE[27] with decreased WHRadjBMI ($P = 1.21 \times 10^{-7}$, Beta = −0.09 SD) in men only (Supplementary Figs. 2, 3, Supplementary Table 3). Associations specific to women were identified for ABCA1, encoding a transporter mutated in Tangier disease[28], and SLC35F5, encoding a protein of unknown function (Supplementary Figs. 2, 4, Supplementary Table 4). The effect of INHBE pLOF on WHRadjBMI was similar between men ($P = 3.87 \times 10^{-4}$, Beta = −0.21 SD) and women ($P = 3.06 \times 10^{-5}$, Beta = −0.22 SD; $P_{het} = 0.87$). In contrast, significantly stronger effects were seen in women for PDE3B pLOF ($P_{het} = 1.58 \times 10^{-8}$; Beta = −0.42 SD in women and −0.06 SD in men), PLIN4 pLOF ($P_{het} = 1.22 \times 10^{-5}$; Beta = 0.21 SD in women and 0.02 SD in men) and INSR pLOF ($P_{het} = 4.50 \times 10^{-8}$; Beta = −0.88 SD in women and 0.05 SD in men) (Supplementary Table 5).

We performed replication analysis of the associations with WHRadjBMI identified via the UKB using exome sequencing data for up to 27,380 individuals from the AMP-T2D-GENES consortium[29]. The most significant association in this analysis was of INHBE pLOF with decreased WHRadjBMI ($P = 9.41 \times 10^{-4}$, Beta = −1.03 SD). PLIN1 pLOF associated with decreased WHRadjBMI with nominal significance ($P = 0.05$, Beta = −0.13 SD). Associations for 9 out of 10 genes tested were directionally consistent with those in our primary analysis (Supplementary Table 6). Meta-analysis of the UKB and AMP-T2D-GENES results increased the significance of 7 of these including INHBE pLOF, PLIN1 pLOF and PLIN4 pLOF (Supplementary Data 3). The effect of INHBE pLOF on WHRadjBMI was significantly larger in T2D-GENES compared to UKB (Beta = −1.03 SD vs −0.23 SD, $P_{het} = 0.01$) which may be due to differences between a disease-centric cohort such as T2D-GENES and a population-based cohort such as UKB, or due to differences in the individual pLOF variants present in each cohort (see below).

To replicate our associations in an additional independent cohort, we examined 29,876 individuals in UKB identifying as White who were not included in our original analysis. Testing the genes identified in our discovery cohort, we found significant associations ($P \leq 0.003$, correcting for 11 genes tested) for variants in three genes: PLIN1, PLIN4, and KEAP1, and a consistent direction of effect for a total of 10 out of 11 genes including INHBE (Supplementary Table 7).

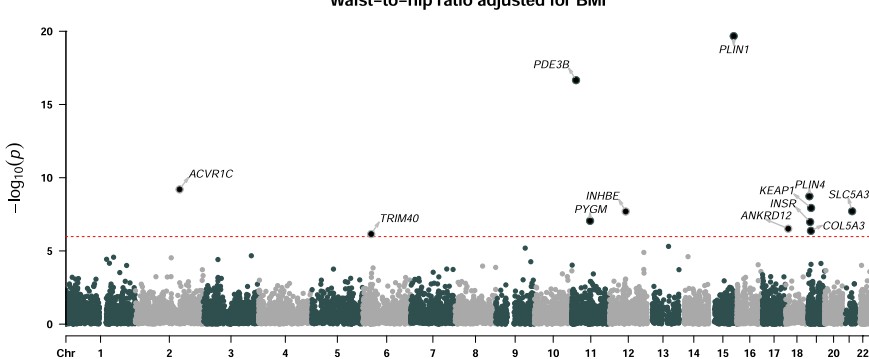

**Fig. 1 Gene-level associations with waist-to-hip ratio adjusted for BMI.** Gene-based burden analysis of WHRadjBMI in 362,679 European ancestry individuals. Association testing was performed using a generalized linear model adjusting for the first 30 principal components of genetic ancestry. The best variant set per gene is shown and significant genes are labeled ($P \leq 1.05 \times 10^{-6}$; Bonferroni correcting for the number of genes and variant masks tested). The dashed line indicates the threshold for statistical significance.

| Table 1 Associations with WHRadjBMI in the exome-wide gene burden analysis. | | | | | |
|---|---|---|---|---|---|
| **Gene** | **Variant set** | **Gene coordinates (hg38)** | **P** | **Beta (95% CI) in SD units of WHRadjBMI** | **N (carriers\|non-carriers)** |
| *PDE3B* | pLOF | 11:14643691-14874139 | $2.17 \times 10^{-17}$ | −0.26 (−0.32, −0.2) | 1020\|361659 |
| *PYGM* | pLOF | 11:64746389-64760715 | $8.75 \times 10^{-8}$ | 0.09 (0.06, 0.13) | 3363\|359316 |
| *INHBE* | pLOF+missense | 12:57455291-57458013 | $2.01 \times 10^{-8}$ | −0.18 (−0.25, −0.12) | 914\|361765 |
| *INHBE* | pLOF | 12:57455291-57458013 | $4.98 \times 10^{-8}$ | −0.22 (−0.30, −0.14) | 618\|362061 |
| *PLIN1* | pLOF | 15:89664365-89679417 | $2.12 \times 10^{-20}$ | −0.35 (−0.43, −0.28) | 681\|361998 |
| *PLIN1* | pLOF+missense | 15:89664365-89679417 | $4.82 \times 10^{-14}$ | −0.11 (−0.14, −0.081) | 4720\|357959 |
| *ANKRD12* | pLOF | 18:9136753-9285985 | $3.02 \times 10^{-7}$ | 0.31 (0.19, 0.43) | 261\|362418 |
| *PLIN4* | pLOF | 19:4502180-4520285 | $1.84 \times 10^{-9}$ | 0.13 (0.091, 0.18) | 1961\|360718 |
| *PLIN4* | pLOF+missense | 19:4502180-4520285 | $1.97 \times 10^{-9}$ | 0.13 (0.086, 0.17) | 2199\|360480 |
| *INSR* | pLOF | 19:7112255-7294405 | $1.05 \times 10^{-7}$ | −0.45 (−0.62, −0.29) | 135\|362544 |
| *COL5A3* | pLOF | 19:9959561-10010532 | $4.20 \times 10^{-7}$ | 0.23 (0.14, 0.32) | 479\|362200 |
| *KEAP1* | pLOF+missense | 19:10486120-10503378 | $1.16 \times 10^{-8}$ | 0.25 (0.16, 0.33) | 520\|362159 |
| *KEAP1* | missense | 19:10486120-10503378 | $2.58 \times 10^{-8}$ | 0.25 (0.16, 0.34) | 477\|362202 |
| *ACVR1C* | missense | 2:157526767-157628887 | $6.24 \times 10^{-10}$ | −0.14 (−0.19, −0.097) | 1892\|360787 |
| *ACVR1C* | pLOF+missense | 2:157526767-157628887 | $7.10 \times 10^{-10}$ | −0.14 (−0.18, −0.095) | 1923\|360756 |
| *SLC5A3* | missense | 21:34073523-34106262 | $1.93 \times 10^{-8}$ | 0.072 (0.047, 0.097) | 6141\|356538 |
| *TRIM40* | pLOF+missense | 6:30135998-30148773 | $6.99 \times 10^{-7}$ | 0.074 (0.045, 0.10) | 4443\|358236 |

Association testing was performed in 362,679 European ancestry individuals using a generalized linear model adjusting for the first 30 principal components of genetic ancestry. Variant sets significantly associating with WHRadjBMI are shown ($P \leq 1.05 \times 10^{-6}$; Bonferroni correcting for the number of genes and variant masks tested).

**Identifying variants contributing to the gene burden associations.** To identify variants contributing to the gene burden associations we performed leave-one-variant-out analysis where we excluded one variant at a time from each variant set and examined how this affected statistical significance. This revealed that the most common pLOF variants often drove the associations seen in the burden tests. For *INHBE*, the most common pLOF variant was present in 538 out of 618 total carriers of pLOF variants and affected a splice acceptor site (rs150777893; NM_031479.4:c.299-1 G > C). This splice acceptor variant contributed most of the signal in the burden test ($P = 0.34$ when rs150777893 was excluded). However, the association was also less significant when the next most common variant, splice donor variant rs375342858, was excluded ($P = 2.44 \times 10^{-7}$ compared to $P = 4.98 \times 10^{-8}$). In single variant tests, rs150777893 significantly associated with decreased WHRadjBMI ($P = 4.31 \times 10^{-8}$, Beta = −0.23 SD) (Supplementary Table 8 and Supplementary Fig. 5). In the AMP-T2D-GENES analysis, an additional *INHBE* variant, rs146517777 (p.Tyr253Ter), was identified that contributes to the association of *INHBE* pLOF with lower WHRadjBMI. Excluding rs146517777 from the analysis resulted in a substantial reduction in significance ($P = 0.08$, Beta =

−0.77 SD compared to $P = 9.41 \times 10^{-4}$, Beta = −1.03 SD when all variants were included) suggesting that this variant has a large impact on WHRadjBMI (Supplementary Table 9). This difference in the alleles carried by participants in AMP-T2D-GENES compared to UKB may contribute to the differences in effect size observed between the two studies. For *PDE3B* and *PLIN1*, the most common pLOF variants (rs150090666 in *PDE3B*; rs750619494 and 15:89667182:C:T in *PLIN1*) also drove the burden associations. In contrast, multiple pLOF variants in *ANKRD12* and *COL5A3* contributed to the gene-based associations. For *ACVR1C*, the missense variant rs56188432 (p.Ile195Thr) was responsible for the association, consistent with published work[9] and, for *SLC5A3*, a previously reported missense variant rs35707420 (p.V370M)[22] drove most, but not all, of the signal (Supplementary Data 4 and Supplementary Fig. 5).

**Relationship between WHRadjBMI genes, abdominal fat and cardiometabolic traits.** Exome-wide gene burden analysis of BMI indicates that genes associating with WHRadjBMI are distinct from those associating with BMI, demonstrating that WHRadjBMI reflects fat distribution rather than overall adiposity

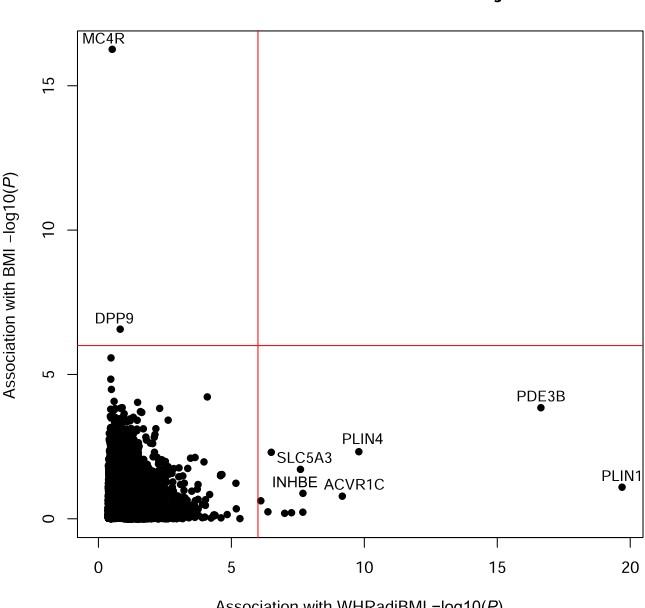

**Associations with BMI and WHRadjBMI**

**Fig. 2 Comparison of associations for WHRadjBMI and BMI.** Results of gene-based association tests for WHRadjBMI and BMI in 362,679 European ancestry individuals performed using a generalized linear model. $-\log_{10}(P)$ is shown for the most significant variant set per gene. The red lines indicate the threshold for statistical significance ($P \leq 1.05 \times 10^{-6}$; Bonferroni correcting for the number of genes and variant masks tested) and selected genes are labeled.

(Fig. 2). This was further supported by examining the association of our WHRadjBMI- associated genes with both unadjusted WHR and BMI. All of these genes associated with WHR but just two genes, *PDE3B* and *PLIN1*, associated with BMI ($P < 0.0016$, adjusting for 16 variant sets and 2 phenotypes tested) although additional genes (*PLIN4, ANKRD12* and *SLC5A3*) reached nominal significance for BMI (Supplementary Fig. 6 and Supplementary Table 10). Consistent with an effect on fat distribution, most genes associated with WHRadjBMI (including *INHBE*) showed a positive relationship between WHRadjBMI and visceral adipose tissue and abdominal subcutaneous adipose tissue as assessed by abdominal MRI (Supplementary Fig. 7). There is an established causal relationship between WHRadjBMI and T2D and CHD[1,3]. Assessing the effects on WHRadjBMI as a function of disease risk revealed that most genes had estimated effects on T2D and CHD that were proportional to the effect on WHRadjBMI based on estimates from MR[1] (Fig. 3). *INSR* pLOF differed from expectations as it associates with lower WHRadjBMI but shows a trend towards increased risk of T2D ($P = 0.08$, OR = 1.70, 95% CI 0.93 to 3.10), consistent with reports from Mendelian genetics[17,18,20].

As abdominal obesity is the most prevalent manifestation of metabolic syndrome (MetS)[6], we examined the association of WHRadjBMI-associated genes with MetS. We identified individuals with MetS traits (i.e., meeting particular thresholds for various biomarkers; Methods)[30] based on UKB baseline assessment data and created a MetS score ranging from 0 to 5 of these traits. Seven out of 11 genes examined associated with MetS score at nominal significance in an ordinal regression including *INHBE* pLOF ($P = 0.02$) and *ACVR1C* missense variants ($P = 0.004$). Strong associations were seen for *PLIN1* pLOF ($P = 1.36 \times 10^{-8}$) and *PDE3B* pLOF ($P = 5.81 \times 10^{-8}$) variants, which is likely driven by their large effect on triglycerides as well as WHRadjBMI (Supplementary Table 11).

**a**

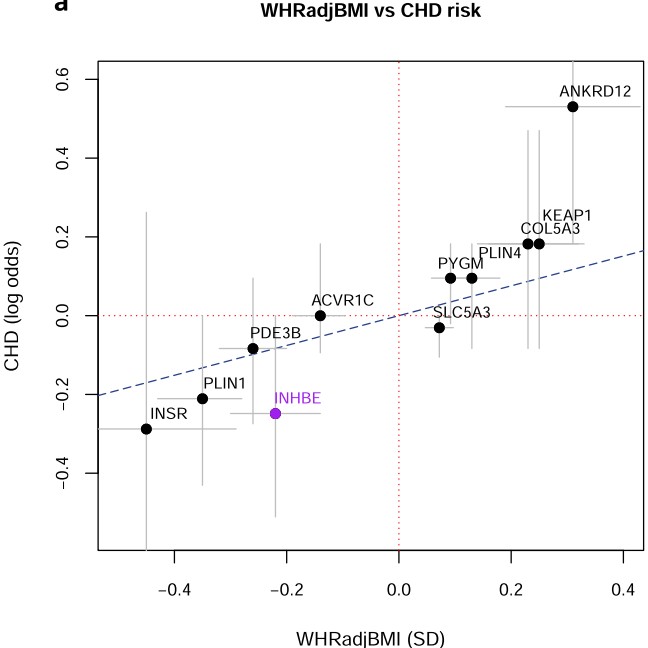

**WHRadjBMI vs CHD risk**

**b**

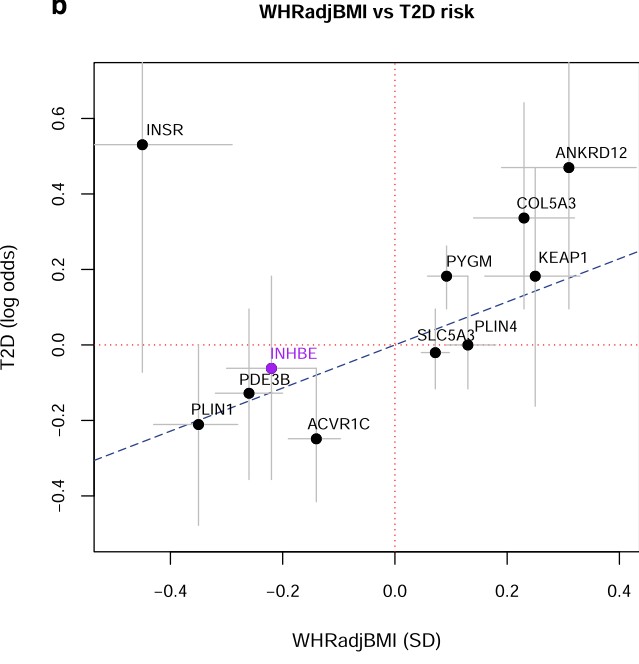

**WHRadjBMI vs T2D risk**

**Fig. 3 Relationship between effect on WHRadjBMI and risk of CHD and T2D.** For significant genes, we plotted the estimated log odds of disease risk as a function of the estimated effect in standard deviations (SD) on WHRadjBMI calculated in 362,679 European ancestry participants. Effects are shown for *INHBE* pLOF and, for the other genes, the most significant variant set per gene. **a** Effect on WHRadjBMI (in SD) versus the log odds of CHD (**b**) Effect on WHRadjBMI (in SD) versus the log odds of T2D. Grey bars represent the 95% confidence interval. The blue dotted line represents the estimated effect on disease predicted by a MR study of WHRadjBMI[1].

**Phenotypic assessment of *INHBE* pLOF carriers**. Genes where pLOF associates with lower WHRadjBMI are of interest as potential therapeutic targets for abdominal obesity and metabolic syndrome. *INHBE* was one such gene identified in our analysis. Heterozygous carriers of *INHBE* pLOF variants (1 in 587

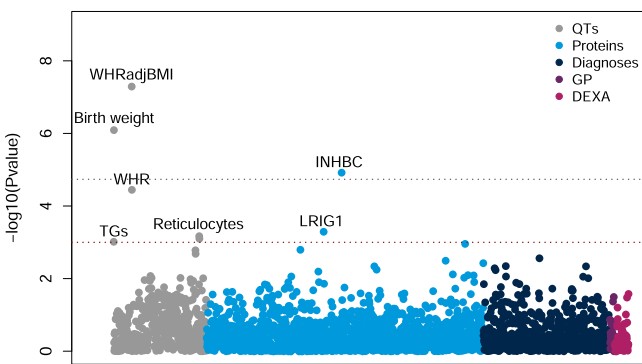

**Fig. 4 PheWAS of *INHBE* pLOF.** The association of *INHBE* pLOF with quantitative traits (QTs), circulating proteins (Proteins), disease diagnoses (Diagnoses), biomarkers from primary care (GP) and measures of body composition (DEXA) was tested using either a generalized linear regression or a mixed-effects model. The y-axis shows $-\log10(P)$ for each trait tested. Phenome-wide significant ($P \leq 1.84 \times 10^{-5}$ Bonferroni correcting for the number of phenotypes tested; grey line) and suggestive ($P \leq 1 \times 10^{-3}$; red line) associations are labeled. TGs; triglycerides, Reticulocytes; high light scatter reticulocyte count and percentage, INHBC; inhibin βC/activin C, LRIG1; leucine-rich repeats and immunoglobulin-like domains protein 1.

individuals, combined frequency of 0.08%) had a favorable metabolic profile consisting of nominally significant associations with lower triglycerides ($P = 9.65 \times 10^{-4}$, Beta = −0.13 SD), higher HDL cholesterol ($P = 0.01$, Beta = 0.10 SD), decreased alanine aminotransferase ($P = 0.04$, Beta = −0.08 SD) and lower fasting glucose ($P = 0.03$, Beta = −0.17 SD). We also detected a non-significant trend towards decreased LDL cholesterol, decreased apolipoprotein B and decreased blood pressure in carriers. *INHBE* pLOF did not associate with BMI but carriers tended to have less visceral and abdominal subcutaneous adipose tissue than non-carriers, supportive of a role for *INHBE* in regulating fat distribution. *INHBE* pLOF associated with decreased WHR without BMI adjustment ($P = 3.57 \times 10^{-5}$, Beta = −0.12 SD) and carriers had a non-significant trend towards decreased waist circumference (Supplementary Table 12).

MR studies have established a causal relationship between WHRadjBMI and cardiometabolic disease risk[1,3]. Consistent with this, we see fewer cases of CHD ($P = 0.05$, OR = 0.78, 95% CI 0.60 to 1.00) and T2D ($P = 0.65$, OR = 0.94, 95% CI 0.70 to 1.24) for *INHBE* pLOF carriers compared to non-carriers which, notably, are proportional to the effect on WHRadjBMI based on estimates from MR[1] (Fig. 3). There were fewer cases of T2D among *INHBE* pLOF carriers in a larger meta-analysis of ~900,000 people but this also did not reach statistical significance (rs150777893 variant; $P = 0.22$, OR = 0.65, 95% CI 0.33 to 1.29)[31]. Given the rarity of *INHBE* pLOF variants, we estimate that we would need to sequence 5-7 million individuals to reliably detect the expected associations with T2D and CHD at $P \leq 0.05$ (Supplementary Table 13).

We performed an in-depth phenome-wide association study (PheWAS) of *INHBE* pLOF to better understand the biological consequences of *INHBE* silencing and to explore any potential safety issues. We tested association of *INHBE* pLOF with 492 quantitative traits including NMR-derived metabolites, 1463 circulating proteins measured using Olink technology, 669 disease diagnoses, a set of 26 clinical measurements present only in UKB primary care data and 72 body composition measurements derived from DEXA imaging and bioelectrical impedance (Fig. 4, Table 2, and Supplementary Data 5–9). This revealed phenome-wide significant associations of *INHBE* pLOF with

increased reported birth weight ($P = 8.09 \times 10^{-7}$, Beta = 0.26 SD) and increased levels of the related protein INHBC in circulation ($P = 1.19 \times 10^{-5}$, Beta = 0.51 SD). We also detected suggestive associations with decreased high light scatter reticulocyte count and increased levels of the protein LRIG1 (Table 2). Notably, the association of *INHBE* pLOF with increased circulating INHBC levels replicated in an independent dataset where proteins were measured using a different technology ($P = 0.006$, Beta = 0.52 SD) (Supplementary Table 14)[32]. We did not detect any associations with body composition beyond WHR.

We examined whether there was any evidence for excess mortality amongst *INHBE* pLOF carriers in a number of ways. Firstly, we performed a survival analysis of *INHBE* pLOF carriers using Cox proportional hazards regression in the UKB and found no significant association of *INHBE* genotype with time to death ($P = 0.16$, Hazard ratio comparing pLOF carriers to non-carriers = 0.79, 95% CI 0.57 to 1.10) (Supplementary Fig. 8). We also looked at *INHBE* pLOF carriers in the gnomAD database (v2.2.1) which has age data for a subset of the exome-sequenced participants. Variants that cause earlier death would be expected to show a trend of lower ages among variant carriers. Of 85,462 exome-sequencing participants with age data available, 72 carried pLOF variants in *INHBE*. Their age distribution was not detectably different than the background (Wilcoxon $P = 0.40$) (Supplementary Data 10). Lastly, we examined whether any of the whole exome-sequenced (WES) or whole-genome sequenced (WGS) individuals in UKB had copy number variants (CNVs) that overlapped or otherwise impacted *INHBE*. Three individuals were identified as having WES-based high-confidence CNVs (all deletions) that either fully or partially overlapped *INHBE*. Two additional individuals were identified based on the WGS data as having the same relatively small deletion that removes most of the second exon of *INHBE* (Supplementary Table 15). All five of the individuals with *INHBE*-overlapping deletions are alive per the latest UKB data release, with ages ranging from 57 to 81 years.

*INHBE* encodes inhibin βE subunit which dimerizes to form activin E, a hepatokine whose signaling is not well-characterized. Notably, the WHRadjBMI-associated gene *ACVR1C* encodes the activin receptor ALK7, further supporting a role for activin signaling in regulating adipose distribution. PheWAS of *ACVR1C* missense variants revealed phenotypic similarities to *INHBE* pLOF including significant associations with birth weight ($P = 1.92 \times 10^{-23}$, Beta = 0.30 SD) and unadjusted WHR ($P = 1.24 \times 10^{-5}$, Beta = −0.08 SD) as well as nominally significant associations with decreased visceral adipose tissue and decreased blood pressure. We also detected the previously reported association with T2D ($P = 0.005$; OR = 0.78) (Supplementary Table 16). *ACVR1C* pLOF variant carriers were rare in our data with just 31 carriers in the European ancestry subpopulation but recent analysis suggests that *ACVR1C* I195T, which drives the WHRadjBMI signal, disrupts ALK7 receptor function[16] further suggesting that inhibiting activin signaling has beneficial effects on fat distribution.

**In vitro characterization of *INHBE* pLOF variants and *INHBE* expression in obesity.** The inhibin βE subunit is a pro-protein consisting of a propeptide domain and a mature domain. Based on what is known about activins A and B, it is likely that inhibin βE pro-proteins associate to form dimers and that the propeptide domain is later cleaved to form activin E[33]. We expressed C-terminus FLAG-tagged inhibin βE in HEK293T cells, which do not endogenously express *INHBE*, and observed the pro-protein present in cell lysate and the mature form in the cell media. We then used this system to characterize the most common *INHBE* pLOF variants in UKB, the splice acceptor rs150777893

**Table 2 PheWAS of *INHBE* pLOF.**

| Title | Variant set | P | Beta in SD (95% CI) | N carrier measured |
|---|---|---|---|---|
| WHRadjBMI | *INHBE* pLOF | $4.98 \times 10^{-8}$ | −0.22 (−0.30, −0.14) | 618 |
| Birth weight | *INHBE* pLOF | $8.09 \times 10^{-7}$ | 0.26 (0.16, 0.37) | 345 |
| WHR | *INHBE* pLOF | $3.57 \times 10^{-5}$ | −0.12 (−0.18, −0.07) | 619 |
| High light scatter reticulocyte percentage | *INHBE* pLOF | $6.74 \times 10^{-4}$ | −0.14 (−0.22, −0.06) | 591 |
| High light scatter reticulocyte count | *INHBE* pLOF | $7.92 \times 10^{-4}$ | −0.14 (−0.22, −0.06) | 591 |
| Triglycerides | *INHBE* pLOF | $9.65 \times 10^{-4}$ | −0.13 (−0.21, −0.05) | 594 |
| INHBC protein | *INHBE* pLOF | $1.19 \times 10^{-5}$ | 0.51 (0.28, 0.74) | 72 |
| LRIG1 protein | *INHBE* pLOF | $5.07 \times 10^{-4}$ | 0.40 (0.17, 0.63) | 72 |

*INHBE* pLOF was tested for association with 492 quantitative traits, 1463 circulating proteins, 669 disease diagnoses, 26 biomarkers from primary care and 72 measures of body composition. Phenome-wide significant ($P \leq 1.84 \times 10^{-5}$; Bonferroni correcting for the number of phenotypes tested) and suggestive ($P \leq 1 \times 10^{-3}$) associations are shown. Full PheWAS results are given in Supplementary Data 5–9.

(NM_031479.4:c.299-1 G > C) and the splice donor rs375342858 (NM_031479.4:c.298 + 1 G > T), as well as rs146517777 (p.Tyr253Ter) found in the AMP-T2D-GENES analysis (Fig. 5a). No inhibin βE protein was made from constructs expressing the splice donor or Tyr253Ter variants, suggesting that these variants are detrimental to protein synthesis (Figs. 5b, c). In contrast, inhibin βE was produced by cells expressing the splice acceptor variant and detected in cell lysate but this protein was present at substantially reduced levels in the cell media (Fig. 5d). Quantification of this protein showed that levels were, on average, 11% ($P = 0.0065$; Fig. 5e) of those produced in cells expressing non-variant *INHBE*, indicating a defect in either protein secretion or stability caused by the splice acceptor variant. This is consistent with a depletion of functional inhibin βE/activin E leading to a lower WHRadjBMI. Sequencing of the mRNA produced by the *INHBE* construct harboring the splice acceptor variant revealed the use of a cryptic splice acceptor site downstream of the variant (AG site at position 309 in NM_031479.4) and an in-frame deletion of four amino acids (NP_113667.1 amino acids 100 to 103; Fig. 5f). These hydrophilic amino acids, Aspartate-Serine-Threonine-Serine, are conserved across mammalian species suggesting functional importance (Supplementary Fig. 9). The fact that the Tyr253Ter variant has a more dramatic effect on inhibin βE protein than the splice acceptor variant may partly account for the larger effect size seen in AMP-T2D-GENES (where this variant was identified) compared to UKB where the splice acceptor variant predominates.

As reduced levels of *INHBE* associate with a healthier fat distribution we asked whether, conversely, *INHBE* is upregulated in conditions of obesity and insulin resistance. We examined *INHBE* expression in the livers of obese monkeys with nonalcoholic steatohepatitis and in the livers of younger lean monkeys. *INHBE* expression was, on average, 3.2-fold higher in the obese monkeys compared to the lean monkeys ($P = 1 \times 10^{-12}$) (Supplementary Fig. 10). While we cannot exclude the possibility that the differences in the age of the monkeys contribute to expression differences, our observations are consistent with previous reports of elevated *INHBE* expression in conditions of insulin resistance[34,35].

**Insights into other WHRadjBMI-associated genes**. Our analysis provided insights into Mendelian lipodystrophy genes and identified additional genes associated with WHRadjBMI. Of note, was the association between *PLIN1* pLOF and decreased WHRadjBMI ($P = 2.12 \times 10^{-20}$, Beta = −0.35 SD). *PLIN1* variants resulting in a longer protein containing 158 aberrant amino acids (frameshift variants rs1567075176 and rs1567075667) have been shown to cause familial partial lipodystrophy type 4 (FPLD4), characterized by a loss of subcutaneous adipose tissue particularly in the gluteal region and lower limbs, hypertriglyceridemia, hypertension, and

type 2 diabetes[23]. The *PLIN1* pLOF variants examined in this study are different to those that cause FPLD4 and are bioinformatically predicted to result in mRNA decay and therefore a complete loss of protein[36]. We see a significant association of *PLIN1* pLOF with decreased triglycerides ($P = 9.56 \times 10^{-12}$, Beta = −0.26 SD), a nominally significant association with decreased blood pressure ($P = 0.02$ for diastolic blood pressure, $P = 0.03$ for systolic blood pressure) and a lower odds ratio for T2D ($P = 0.11$, OR = 0.81, 95% CI 0.62 to 1.05) opposite to the lipodystrophy phenotype. We also observed significant associations with HDL cholesterol, hip circumference, and reticulocyte count (Supplementary Table 17).

We specifically examined WHRadjBMI associations for 13 genes implicated in Mendelian lipodystrophies in both our sex-combined and sex-stratified analyses. We identified significant associations for 4 out of these 13 genes, *PLIN1*, *LIPE*, *LMNA* and *PPARG*, ($P \leq 0.004$, correcting for 13 genes tested) in at least one analysis. Rare variants in an additional three genes, *CAV1*, *AGPAT2* and *CAVIN1*, associated with WHRadjBMI at nominal significance (Supplementary Fig. 11 and Supplementary Data 11).

PheWAS of the other WHRadjBMI-associated genes provided further biological insights. For example, *PLIN4* pLOF (WHRadjBMI $P = 1.84 \times 10^{-9}$, Beta = 0.13 SD) was associated with a decrease in overall body fat percentage ($P = 7.56 \times 10^{-7}$, Beta = −0.08 SD) suggesting a redistribution of fat to the abdomen in carriers. We observed pleiotropic effects of *ANKRD12* pLOF which associated with a range of diverse traits including total protein, blood cell counts, performance on cognitive tests, and bronchitis. Finally, *SLC5A3* missense variants associated with biomarkers of kidney function and *KEAP1* pLOF + missense associated with creatinine levels, apolipoprotein A and eosinophil counts (Supplementary Data 12).

## Discussion
Our exome-wide analysis of WHRadjBMI, a surrogate for abdominal fat, highlights loss of function in *INHBE* as a genetic factor contributing to a healthier fat distribution. *INHBE* is distinct from many of the other WHRadjBMI-associated genes as it is predominately liver-expressed and encodes a secreted protein which likely exerts its effects in other tissues such as adipose. Characterization of the most common (MAF = 0.08%) *INHBE* pLOF variant showed that the resulting protein was present intracellularly but that levels of secreted protein were reduced by nearly 90%, likely severely compromising its ability to signal to other tissues. *INHBE* encodes inhibin βE subunit, an activin/inhibin component which belongs to the TGF-beta superfamily but whose signaling has not been characterized[33]. Based on other family members[33], we hypothesize that the inhibin βE subunit dimerizes to form activin E. Although receptors for activin E have not been identified, the past observations that variants in

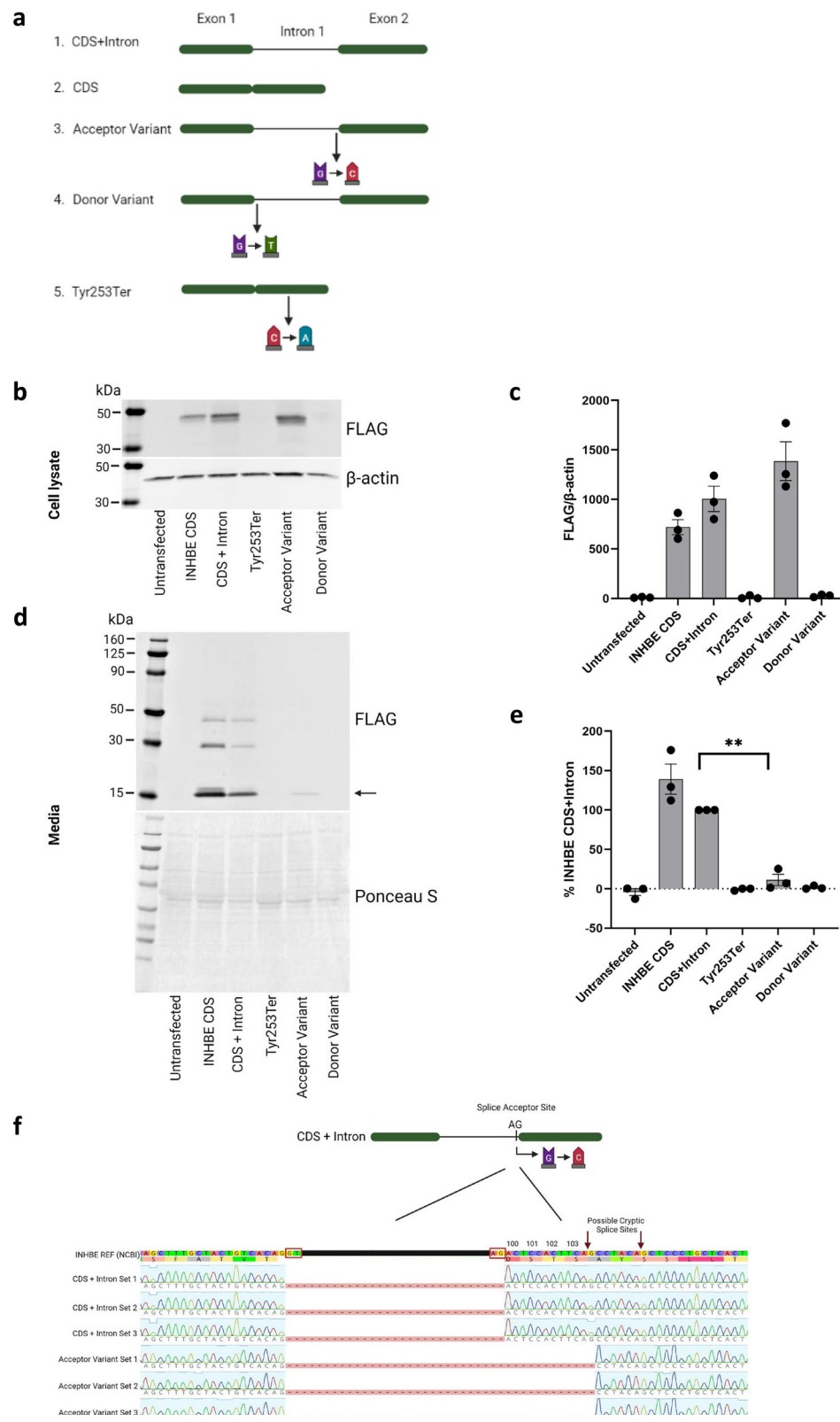

*ACVR1C*, encoding the activin receptor ALK7, associate with WHRadjBMI[9] which were replicated in this study lead us propose ALK7 as a candidate for an activin E receptor.

Further analysis of *INHBE* pLOF carriers revealed a favorable metabolic profile which included decreased triglycerides, increased HDL cholesterol, and decreased fasting glucose. An extensive PheWAS of *INHBE* pLOF did not reveal any associations suggesting adverse effects of *INHBE* pLOF and carriers of these variants did not show evidence of excess mortality. Of note, was the association of *INHBE* pLOF with increased circulating levels of INHBC protein. INHBC dimerizes to form activin C which has recently been shown to signal through the ALK7

**Fig. 5 In vitro characterization of *INHBE* pLOF variants. a** Design of *INHBE* expression constructs containing the splice acceptor variant, the splice donor variant and stop gain variant Tyr253Ter which were transfected into HEK293T cells. CDS; coding sequence (**b**) Western blot analysis of FLAG-tagged inhibin βE protein from cell lysate. The expected size of the FLAG-tagged inhibin βE pro-protein is 39–42 kDa. β-actin was used as a loading control. **c** Quantification of inhibin βE in cell lysate from 3 independent biological replicates, error bars represent the standard error of the mean (SEM). **d** Western blot analysis of secreted FLAG-tagged inhibin βE protein harvested from the media. Ponceau S protein stain was used as a loading control. The arrow represents the expected size of the FLAG-tagged inhibin βE mature domain (14–16 kDa). **e** Quantification of inhibin βE in cell media from 3 independent biological replicates. Values are normalized to non-variant *INHBE* (CDS + intron). Error bars represent SEM, **$P = 0.0065$ using a two-sided unpaired t-test. **f** Sequencing of the mRNA produced by cells expressing non-variant *INHBE* (CDS + intron) and cells expressing the splice acceptor variant. Amino acid residues are numbered, and red boxes indicate splice sites. Set 1, 2 and 3 refer to independent biological replicates. Panels (**a**) and (**f**) were created with BioRender.com.

receptor in mature adipocytes[37]. Whether INHBC is upregulated to compensate for a reduction in INHBE or some other mechanism is at play remains to be investigated.

Notably, several non-genetic studies implicate *INHBE* in metabolic dysfunction. Two studies report a correlation between increased expression of *INHBE* and insulin resistance in both humans and mice[34,35]. Here, we observed higher expression of *INHBE* in the livers of obese monkeys compared to lean monkeys providing further support for this. Hepatic knockdown of *Inhbe* in db/db diabetic mice was found to reduce body fat consistent with our findings from human genetics[35]. However, a different study showed that overexpression of *Inhbe* increased energy expenditure and improved insulin resistance in high fat diet fed mice[38]. These discrepancies may simply reflect the differences between mouse models, which are an imperfect system for studying human adiposity, or may suggest that precise levels of activin E are important for regulating energy metabolism and fat deposition.

Our analysis of rare genetic variation influencing WHRadjBMI identified additional genes with a role in regulating fat distribution and metabolic function including the known Mendelian disease genes, *PYGM*, *INSR*, *PLIN1*, *LIPE* and *ABCA1*. With the exception of *PLIN1*, all genes identified have phenotypic associations reminiscent of the corresponding Mendelian diseases. For example, biallelic loss of function in *PYGM* causes glycogen storage disease which can lead to muscle atrophy and an increase in adipose tissue[24,39]. Recessive mutations in *INSR* cause Donohue syndrome, a feature of which is a lack of adipose tissue[40]. *INSR* variants are also known to cause insulin resistance and diabetes[17,18,20]. Consistent with this, and in contrast to other genes associated with lower WHRadjBMi, we see a trend towards increased T2D risk in *INSR* pLOF carriers. Of note, is our finding that *PLIN1* pLOF associates with lower WHRadjBMI and lower triglyceride levels, which is consistent with previous studies[16,41]. This suggests that therapeutic silencing of *PLIN1* may have beneficial metabolic outcomes which are different to the effects of the mutant perilipin 1 protein seen in FPLD4, an autosomal dominant disorder characterized by loss of adipose tissue in the lower body, T2D, and hypertriglyceridemia[23].

We also identified WHRadjBMI associations for several genes with known links to adipose biology or insulin resistance including *COL5A3* and *PLIN4*. *Col5a3−/−* mice have previously been shown to be glucose intolerant, insulin-resistant and have sex-specific decreases in dermal fat[42]. *PLIN4* encodes perilipin 4 which coats newly synthesized lipid droplets to form a stable protein layer[43,44]. Another identified gene, *KEAP1*, is a negative regulator of NRF2 which plays a role in adipocyte differentiation while *GIGYF1* pLOF, which associated with higher WHRadjBMI in men only, has known T2D associations[45].

Loss of function in *ANKRD12*, encoding a putative transcriptional repressor, had pleiotropic effects which ranged from increased WHRadjBMI to worse performance in several cognitive tests. This may suggest a multi-organ syndrome in pLOF carriers

and, consistent with this, *ANKRD12* is expressed in a broad range of tissues[46]. Several of the genes with rare variant associations, namely *COL5A3, PLIN4*, and *SLC35F5* also have reported common variant associations for WHR[11,22]. However, no previous genetic studies have implicated *INHBE* in the control of adipose distribution in humans.

Our findings from exome-sequencing of over 360,000 individuals highlight *INHBE* as a novel therapeutic target to treat abdominal obesity and cardiometabolic disease. The causal relationship between WHRadjBMI and cardiometabolic disease is well established[1,3]. Consistent with this, carriers of *INHBE* pLOF variants have a more favorable metabolic profile and estimated lower odds of CHD and T2D than non-carriers. Importantly, by reducing abdominal fat, drugs targeting *INHBE* would have a distinct biological mechanism to existing drugs for CHD and T2D and may complement current therapies.

## Methods

**Ethics statement**. The UK Biobank study was approved by the National Health Service National Research Ethics Service and all participants provided written informed consent to participate in the study. The UK Biobank resource is an approved Research Tissue Bank and is registered with the Human Tissue Authority, which means that researchers who wish to use it do not need to seek separate ethics approval (unless re-contact of participants is required). Information about ethics oversight in the UK Biobank can be found at https://www.ukbiobank.ac.uk/ethics/. This research has been conducted using the UK Biobank resource, applications 26041 and 65851. All individuals in the AMP-T2D-GENES study provided informed consent and all samples were approved for use at the respective institution.

Ethical approval for animal experiments was obtained from the Charles River Laboratory Institutional Animal Care and Use Committee (IACUC) and the Kunming Biomed International IACUC.

**The UK Biobank resource**. The UK Biobank (UKBB) recruited ~500,000 participants in England, Wales, and Scotland between 2006 and 2010[47]. Phenotypic data available includes anthropometric traits, biomarker data and self-reported diseases collected at the time of baseline assessment as well as disease diagnoses from inpatient hospital stays, the cancer registry and death records obtained through the NHS. Approximately half of the participants also have diagnoses from primary care available and a subset of participants have undergone abdominal imaging. Array genotypes are available for nearly all participants and exome sequencing data is available for 454,756 participants.

**Exome sequencing, population definition and PC calculation**. DNA was extracted from whole blood and sequenced by Regeneron Genetics Center as described elsewhere[48]. Briefly, the xGen exome capture was used and reads were sequenced using the Illumina NovaSeq 6000 platform. Reads were aligned to the GRCh38 reference genome using BWA-mem[49]. Duplicate reads were identified and excluded using the Picard MarkDuplicates tool (Broad Institute). Variant calling of SNVs and indels was done using the WeCall variant caller (Genomics Plc.) to produce a GVCF for each subject. GVCFs were combined to using the GLnexus joint calling tool[50]. Post-variant calling filtering was applied using the Goldilocks pipeline[48]. Variants were annotated using the Ensembl Variant Effect Predictor (VEP) v95[51] which includes a LOFTEE plug-in to identify high confidence pLOF variants[36]. Combined Annotation Dependent Depletion (CADD) scores[52] were generated using the Whole Genome Sequence Annotator (WGSA) AMI version 0.8. Positions are based on the hg38 genome build.

Subject quality control and determination of genetic relationships between participants were performed by Regeneron Genetics Center (RGC) and removed subjects with evidence of contamination, unresolved duplications, sex discrepancies

and discordance between exome sequencing and genotyping data. Genetic relationships between participants were determined by RGC using the PRIMUS program[53]. Our analysis of WHRadjBMI was performed in the unrelated subset from which all first- and second-degree relatives and some third-degree relatives had been excluded.

Sub-populations were defined through a combination of self-reported ethnicity (Field 21000) and genetic principal components (PCs). PCs were calculated using methodology outlined elsewhere[45]. Briefly, we ran an initial principal component analysis on high quality variants (missingness <2%, MAF > 0.1%, HWE $P \geq 10^{-12}$, pruned to independent markers) using eigenstrat[54]. We then removed all individuals $+/-$ 3 standard deviations from the mean of PCs 1–6. A final PC estimation was then performed using unrelated subjects. We then projected related individuals onto the PCs. For sensitivity analysis in the White outgroup, we selected individuals reporting a White ethnic background (Field 21000) but who were not included in the European ancestry sub-population used in our discovery analysis. In this sensitivity analysis, we adjusted for genetic ancestry using 30 PCs from a set supplied by UKB (Field 22009).

**Calculation of WHRadjBMI and definition of other traits**. We calculated WHRadjBMI for UKB participants using manual measurements for waist circumference (Field 48), hip circumference (Field 49), and BMI (Field 21001) taken at baseline assessment. A linear model was built modeling waist-to-hip (WHR) and adjusting for age at recruitment, sex, and BMI. WHRadjBMI was defined using the residuals from this model similar to previous studies[11] and values were inverse rank normalized using the RNOmni R package[55] prior to association testing. For sex-specific analyses, WHRadjBMI was calculated separately for men and women (defined based on Fields 31 and 22001) excluding sex from the linear model. Inverse rank normalization of WHRadjBMI values was performed separately for men and women.

Imaging-derived measures quantification of visceral adipose tissue and abdominal subcutaneous adipose tissue volume were obtained from UKB (Fields 22407 and 22408), adjusted for BMI as previously described[13], and inverse rank normal transformed for association testing. Liver proton density fat fraction measurements were obtained from UKB (returned dataset 2343)[56] and inverse rank normal transformed for association testing.

For phenome-wide analyses of the WHRadjBMI associated genes, a selection of ~290 quantitative traits was obtained from other fields, encompassing anthropometric measurements, cognitive tests, blood counts, as well as blood and urine biochemistry. For measurements of LDL cholesterol, we also constructed a phenotype that was adjusted for use of cholesterol lowering medication as previously described[57]. All quantitative traits were inverse rank normalized using the RNOmni R package[55] prior to association testing. Disease diagnoses were extracted from inpatient hospital diagnoses, the death and cancer registries, primary care, and self-report. Phecodes were constructed from ICD10 coded-diagnoses[58]. CHD was defined as "phecode 411 ischemic heart disease" and T2D as "phecode 250.2 type 2 diabetes". Diseases with >1000 cases were tested in PheWAS.

For *INHBE* pLOF additional phenotypes were examined. These included an expanded set of quantitative traits ($N = 492$) including NMR metabolites (category 220), circulating proteins quantified using Olink technology ($N = 1463$) as well as body composition measured by impedance (category 100009) and DEXA-imaging (category 124) ($N = 72$). All quantitative traits were inverse rank normalized prior to association testing using the RNOmni R package[55]. In addition, we tested for association with all disease diagnoses with >500 cases ($N = 669$). We also curated a set of clinical measurements which were absent from the UKB baseline assessment data but had > 20,000 individuals with measurements in primary care data (GP clinical events, field 42040) ($N = 26$). These measurements were mapped to the OMOP common data model. For each measurement, values were extracted, and outliers were removed ($+/-4$ SD from the mean). Values were then inverse rank normalized and adjusted for age at measurement using a linear model. For each participant, the mean of the computed residuals was calculated to give a single value per person. All phenotypes tested are listed in Supplementary Data 5–9.

Individuals with metabolic syndrome traits at baseline assessment were identified as follows; waist circumference >89 cm for women and >102 cm for men, triglycerides >1.7 mmol/L, HDL cholesterol <1.30 mmol/L in women and <1.04 mmol/L in men, blood pressure >130/85 mm Hg, and HbA1c > 5.7%. This is consistent with clinical criteria for defining metabolic syndrome[30], although we substituted HbA1c for fasting glucose as those measurements were not available for all participants. Participants were given a metabolic syndrome score between 0 and 5 reflecting the number of metabolic syndrome traits present.

**Association testing in UKB**. For gene-based tests, variant masks included autosomal variants only and were defined as follows: pLOF variants were predicted protein truncating variants (stop gain, frameshift, splice acceptor, splice donor) called as high confidence by LOFTEE, predicted damaging missense variants were missense variants with a CADD PHRED-scaled score ≥25. All included variants had MAF ≤ 1%, missingness across individuals ≤2%, and HWE $P \geq 10^{-10}$. For exome-wide gene burden analysis of WHRadjBMI we employed a $p$-value threshold Bonferroni corrected for the number of genes and variant masks tested ($P \leq 1.05 \times 10^{-6}$).

For quantitative traits, burden testing was performed in unrelated individuals using a generalized linear model implemented in R according to a gaussian model. Genotype was coded according to a dominant model; 0 (no variant) or 1 (any number of variants). When testing associations with WHRadjBMI we adjusted for the first 30 PCs of genetic ancestry in the regression as values had previously been adjusted for age and sex. For other quantitative traits we adjusted for age at recruitment, sex and 30 PCs. For quantitative traits, we required at least 10 rare variant carriers per gene to have measurements in our primary analysis. Case-control analyses were performed using a mixed-effects model implemented in REGENIE v2.2.4[59] and included related individuals. We adjusted for age, sex, the availability of primary care data, country of recruitment and 30 PCs in the regression. For case-control analyses we only tested variant sets where there were at least 5 cases amongst rare variant carriers. For the sensitivity analysis of WHRadjBMI in the White outgroup we tested significant variant sets where we had ≥5 rare variant carriers per gene.

For traits where the effect in clinical units is shown, to convert effect sizes from normalized values back to measured units, the estimates from the regression were multiplied by the standard deviation of these traits in the entire cohort.

Meta-analysis was performed using an inverse-variance weighted method implemented in METAL (version release date 2011-03-25) and heterogeneity was tested using Cochran's Q test[60].

Single variant associations with WHRadjBMI were tested using an additive model in PLINK[61] adjusting for the first 30 PCs of genetic ancestry.

For MetS score, proportional odds logistic regression was performed using "polr" (MASS package) in R. The effect in terms of number of MetS traits was then calculated by running a linear regression.

Conditional analysis was performed by including the relevant genotype as a covariate in the regression. Common variant associations with WHRadjBMI were identified using genotypes from array typing and imputation[62] via PLINK[61] adjusting for genetic ancestry via 30PCs. We adjusted our gene burden associations for the lead common SNP in a 250 kb window with $p \leq 1.05 \times 10^{-6}$.

**Replication analysis in AMP-T2D-GENES dataset**. The AMP-T2D-GENES dataset[29], comprised of exome sequencing data from 20,791 T2D cases and 24,440 controls, was used for replication analysis of variant sets with significant associations in our primary sex-combined analysis. All signals were tested in up to 27,380 unrelated individuals with reported WHR and BMI values. WHR ratio was adjusted for age, sex, and BMI using a linear model and the calculated residuals for each trait were inverse-normal transformed to derive WHRadjBMI. As previously reported[63], transformations were performed separately within each cohort of the AMP-T2D-GENES dataset. Individuals with reported WHRadjBMI data spanned five genetic ancestries: African American ($N = 1411$), East Asian ($N = 5935$), European ($N = 6412$), Hispanic ($N = 9253$), and South Asian ($N = 4369$). Ancestries were confirmed by visual inspection of the first two PCs based on a multi-ancestry analysis of common variants. Variants were annotated using the VEP software package v87[51] using the LOFTEE plugin[36] to identify high confidence pLOF variants and CADD PHRED score[52]. Transcripts were chosen using the "–flag-pick-allele" option implemented in VEP. Variant masks corresponded to those used in our primary analysis pLOF, missense, and pLOF+ missense with all MAF ≤ 1%. Variant masks with fewer than 5 carriers (i.e., *ANKRD12* pLOF) and *TRIM40* pLOF+ missense (due to proximity to HLA gene cluster) were excluded from the replication analysis. Genomic positions are based on the GRCh37/hg19 genome build.

EPACTS software package version 3.2.4 (genome.sph.umich.edu/wiki/EPACTS) was used to perform a burden test with linear regression. As previously done[29], testing was performed as a single "mega-analysis" across all samples. In this method, we used a set of unrelated samples (all pairs IBD < 0.25) and added 10 ancestry PCs, sample cohort subgroup, and sequencing technology as covariates. The variant filtering was performed at the sample subgroup level by setting genotypes for an entire subgroup as missing during association testing as previously described[29].

**Estimation of effects based on MR and power calculations**. MR studies have estimated that for every 1 SD increase in WHRadjBMI the odds ratio of CHD is 1.46 and the odds ratio of T2D is 1.77[1]. We natural log transformed these odds ratios and calculated expected effects over a range of WHRadjBMI effect sizes. We fit a linear model, ln(odds of disease) ~ WHRadjBMI effect in SD, to plot how disease risk is estimated to change for WHRadjBMI-associated variant sets. For *INHBE* pLOF, WHRadjBMI is 0.22 SD lower and the MR-estimated odds ratio is 0.92 for CHD and is 0.88 for T2D. We used the genpwr package in R to calculate the cohort size needed to detect these effects at $P \leq 0.05$, assuming an additive model and a combined MAF of 0.08%.

**Survival analysis of *INHBE* pLOF carriers and assessment of their age distribution in gnomAD**. The association between the *INHBE* pLOF genotype and time to death (category 100093) was tested in UKB using Cox proportional hazards regression starting at date of enrollment and controlling for age at enrollment, sex, and genetic ancestry via 30 principal components and implemented using the

"survival" package in R. Kaplan-Meier curves were used to visualize survival by *INHBE* genotype.

To examine the age distribution of *INHBE* pLOF carriers in gnomAD v2.1.1, the vcf file for exome data from chromosome 12 was obtained from the gnomAD website (https://gnomad.broadinstitute.org/downloads). Variants were filtered to pass QC among the exome calls and to be pLOF in *INHBE* using the LOFTEE filter as implemented in gnomAD. Variant frequency by age (provided in 12 age bins) was extracted from the VCF and summed across variants. The overall age distribution was obtained from the gnomAD variant view page. A Wilcoxon test was performed in R on the variant carriers' ages (encoded as ordinal age bins) vs. non-carriers' ages.

**Identification of CNVs overlapping *INHBE* in UKB WES and WGS data**. We examined whether any of the 454,756 WES samples had CNVs that overlapped or otherwise impacted *INHBE*. We focused on CNVs with a maximum size of 100 Kb, as larger CNVs would impact a large quantity of genes beyond *INHBE* and thus be less informative with respect to potential consequences of *INHBE* copy number variation. CNV calls were made by the Regeneron Genetics Center by applying the CLAMMS pipeline[64] (version 1.3) to the WES data. The CNV call set included 449,872 'inlier' samples (those individuals remaining after removing samples defined as 'outliers' based on having >40 CNVs or >40,000 exons called as a CNV). Among these samples, three individuals were identified as having WES-based high-confidence CNVs (all deletions) that either fully or partially overlapped *INHBE*. High-confidence CNVs were defined as CNVs with a QC score of 2 or 3 given a score range of 0–3, with QC score determined based on a combination of CNV call-level metrics (Q_nondip and Q_exact), SNP information (for deletions, heterozygosity/homozygosity ratio was considered), and performance of the locus across the cohort. In addition to the WES dataset, 150,119 UK Biobank samples have undergone whole genome sequencing (WGS) and deCODE genetics has employed a pipeline involving MANTA[65] (version 1.6) and GraphTyper[66] (version 2.6) to identify CNVs based on these WGS data[67]. Of the three samples with high-confidence deletions overlapping *INHBE* based on WES data, two samples were included amongst the 150,119 WGS samples. The WES-identified deletions for these two samples were also identified based on the WGS data, with slightly different breakpoints as shown in Supplementary Table 15. Furthermore, two additional samples were identified based on the WGS data as having the same high-quality (alternate allele score [AAscore] = 0.978, genotype quality >20 and read depth >10), relatively small (~2 Kb) deletion that partially overlaps *INHBE* (deleting most of the 2nd exon); these samples were present amongst the 'inlier' samples of the WES CNV call set, but these deletions were not identified by the WES CLAMMS pipeline.

**Tissue expression enrichment of WHRadjBMI-associated genes**. For a set of 15,947 protein-coding genes for which RNA expression was available in GTEx v8[46], the mean expression across individuals for each of the 54 tissues was obtained, and the tissue with the highest expression was delineated for each gene. For each tissue, a one-sided Fisher's exact test was performed for enrichment among the genes for which that tissue had the highest expression for the genes associated with WHRadjBMI (defined as $N = 11$ genes with WHRadjBMI $p < 1 \times 10^{-6}$ or $N = 73$ genes with WHRadjBMI $p < 1 \times 10^{-3}$.) In addition to the 54 GTEx tissues, nine additional groups of tissues (adipose, artery, brain, cervix, colon, esophagus, heart, kidney, skin) with multiple GTEx tissues were aggregated and also tested.

**In vitro characterization of *INHBE* variants**. Human *INHBE* sequences were obtained from NCBI synthesized in pCMV6-AC-3DDK (Origene) containing a linker sequence and 3× FLAG at C-terminus by BlueHeron Biotech (Bothwell, WA). The following plasmids were generated: CDS contained the coding region of INHBE, CDS + Intron contained the coding region of INHBE with the intron, INHBE CDS with rs146517777 mutation (Tyr253Ter), INHBE CDS + Intron with rs150777893 (splice acceptor variant), and INHBE CDS + Intron with rs375342858 (splice donor variant). For sequence analysis, the relevant region of the plasmid was amplified using PrimeSTAR Max DNA polymerase master mix (TakaraBio, Kusatsu, Shiga, Japan). The PCR product was purified using QIAquick PCR Purification Kit (Qiagen, Hilden, Germany), followed by Sanger sequencing (Genewiz, Cambridge, MA). All sequence analyses were formed using Geneious Prime 2021.2.2 (Biomatters Ltd, Auckland, New Zealand).

HEK293T cells (CRL-3216 ATCC; Manassas, VA) were grown in Dulbecco's Modified Eagle Medium (DMEM) with 10% Fetal Bovine Serum (FBS; Gibco, Carlsbad, CA) at 37 °C and 5% $CO_2$. All transfections were performed using Lipofectamine™ 3000 (Invitrogen, Carlsbad, CA) and Opti-MEM (Gibco, Grand Island, NY) according to the manufacturer's specifications. 10 ug of each plasmid were transfected into 100 mm tissue culture plates. 24 hrs post-transfection, media was changed to serum-free. Cells and media were collected 48 h post-transfection. Briefly, 10 mL of media was collected from each condition and spun at $1000 \times g$ for 10 mins, the supernatant was collected followed by another spin. Halt protease and phosphatase inhibitor cocktail was added to the resulting supernatant, followed by ~40-fold concentration of the media using Pierce™ Protein Concentrator PES with a molecular weight cutoff of 3 kDa (ThermoFisher Scientific, Waltham, MA).

Following media removal, the transfected plates were washed in 1× PBS followed by brief trypsinization and resuspension in DMEM. Cells were pelleted at $300 \times g$ for 5 mins and the supernatant was discarded. Cell pellets were immediately frozen on dry ice.

Transfected cell pellets were lysed on ice in 1× RIPA buffer containing Halt Protease and Phosphatase inhibitor. Lysis was performed with intermittent vortexing for 30 mins, followed by sonication at power setting 3 (Microson Ultrasonic Cell Disruptor XL (Misonix, Farmingdale, NY). Sonication was performed on ice for 3 cycles of 10 s, with a 30 s cool-off. The resulting lysate was spun at 14,000 g for 15 mins at 4 °C. The supernatant was collected and used for western blot analysis. Total protein quantification was performed using the Pierce™ BCA Protein Assay Kit (ThermoFisher Scientific, Waltham, MA). 18 ug of the total lysate was loaded on to a Novex 4–20% Tris-Glycine gel. For secreted media analysis, a standard volume of 20 ul was loaded. Gels were transferred on to PVDF membranes using iBlot™ 2 Transfer Stacks (ThermoFisher Scientific, Waltham, MA), followed by blocking in 5% nonfat dry milk (NFDM) in 1×TBS with 0.05% Tween. Anti-FLAG antibody (1:6000, MilliporeSigma, Burlington, MA) was added for overnight incubation with rotation at 4 °C, followed by incubation with IRDye® 680RD Goat anti-Mouse IgG Secondary Antibody (LI-COR, Lincoln, NE) at 1:10,000 in 5% NFDM with 1xTBST for 1 h at room temperature. Anti-βactin (1:8000, MilliporeSigma, Burlington, MA) was used as loading control. The blots were imaged using Chemidoc MP (Biorad, Hercules, CA) and Odyssey (LI-COR Lincoln, NE), followed by densitometry analysis using ImageStudio (LI-COR Lincoln, NE). For secreted media blots, protein loading was assessed by Ponceau S (MilliporeSigma, Burlington, MA). Images were prepared for publication using BioRender (biorender.com, Toronto, Ontario). Three independent biological replicates were performed.

**Expression analysis of *INHBE* in lean and obese cynomolgus monkeys**. Liver biopsies were taken from 24 lean cynomolgus monkeys (age 2–4 years, weight 1.9–6.0 kg, male and female) and 13 aged obese monkeys on a high fat diet displaying features of non-alcoholic steatohepatitis (age >8 years, weight >7.0 kg, male). RNA was extracted from liver tissue by TRIzol-chloroform extraction and purified with Qiagen RNeasy 96 Universal Tissue Kit. cDNA was generated using Applied Biosystems High-Capacity cDNA Reverse Transcription Kit. Taqman probe-based qPCR was used to quantify *INHBE* mRNA (Mf02820386_g1, ThermoFisher) which was normalized to the geometric mean of two housekeeping genes, *ARL6IP4* (Mf02792752_g1, ThermoFisher) and *RPS9* (Mf04389309_m1, ThermoFisher).

**Reporting summary**. Further information on research design is available in the Nature Research Reporting Summary linked to this article.

## Data availability

For the primary analysis in UKB, phenotypic data, exome sequencing and whole genome sequencing data are accessible through application to UKB with the exception of proteomics data which is scheduled for release in October 2022. For replication analysis in AMP-T2D-GENES, details on how to access results from the individual studies can be found in the original publications[29,68–71] and a list of dbGAP/EGA accession numbers is provided in Supplementary Data 13. Additional data is publicly available through the T2D Knowledge Portal at https://t2d.hugeamp.org/dinspector.html?dataset=ExSeq_52kQT. gnOMAD data is available for download at https://gnomad.broadinstitute.org/downloads. Source data are provided with this paper.

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

## Acknowledgements
This research has been conducted using the UK Biobank Resource (Applications 26041 and 65851). We would like to thank the participants and researchers of UK Biobank for creating an open-access resource. We thank the UK Biobank Exome Sequencing Consortium and UK Biobank for facilitating exome sequencing of participants and the UKB Pharma Proteomics Project for generating proteomics data. We also thank individuals from the cohorts analyzed by the AMP-T2D-GENES consortium. We thank Dan Berman for help with data analysis, Ho-Chou Tu, Lucas BonDurant, Kristina Yucius and Anna Borodovsky for providing samples, and Josh Friedman for critical feedback. We thank Megha Subramanian and Kirsten Deprey for reagents. Data management and analytics were performed using the REVEAL/SciDB translational analytics platform from Paradigm4. J.F. and P.D. were supported by R01DK125490.

## Author contributions
A.Deaton, L.D.W., M.M.P., R.A.H., C.W., M.E.P. and A.M.H. performed computational analyses; P.D. and J.F. performed replication analysis in AMP-T2D-GENES; A.Dubey, E.Y., S.T. and L.N. designed and carried out experiments; A.Deaton, A.Dubey, L.D.W., P.D., J.F., S.T., G.H., K.F., A.K.V and P.N. interpreted results. A.Deaton wrote the manuscript. All authors reviewed and edited the manuscript.

## Competing interests
A.Deaton, A.Dubey, L.D.W., E.Y., S.T., L.N., R.A.H., C.W., M.E.P., A.M.H., K.F., A.K.V. and P.N. are employees and stockholders of Alnylam Pharmaceuticals. G.H. is an employee and stockholder of Alnylam Pharmaceuticals and a paid consultant to 54Gene. M.M.P. is a former employee and stockholder of Alnylam Pharmaceuticals and a current employee and stockholder of Beam Therapeutics. As of April 2022, P.D. is an employee and stockholder of Regeneron Pharmaceuticals. J.F. has no competing interests.

## Additional information

## AMP-T2D-GENES Consortium

Peter Dornbos[2,3,4] & Jason Flannick [iD][2,3,4]

A full list of members and their affiliations appears in the Supplementary Information.

