## [Peer Review File · Nature Communications]

Rare loss of function variants in the hepatokine gene INHBE protect from abdominal obesityREVIEWER COMMENTS

Reviewer #1 (Remarks to the Author):

Deaton and colleagues present an interesting analysis of body fat distribution in UK Biobank. Using the whole exome sequences from UK Biobank they show that loss-of-function variation in INHBE is associated with reduced WHRadjBMI. They further perform in vitro characterization of the lead splice variant in INHBE and show that it associates with a 90% reduction in secreted INHBE. Functional validation of two additional pLOF variants showed no intracellular expression.

These findings identify INHBE as a genetically-validated target for body fat distribution and related disorders, including lipodystrophy and potentially T2D & CHD. I have only minor comments on this very interesting study.

1. Any thoughts on why the T2D Genes effect of INHBE LOF is five times as large as the UKBB effect? There appears to be significant heterogeneity between the two estimates. Different variants?
2. It would be worthwhile showing the effects of the hits on WHR and BMI independently, given that conditioning on BMI will select for variants which increase or decrease BMI.
3. I would take a look at glucose (available in UKBB) as well as HbA1c to see if INHBE deficiency affects glycaemic traits.
4. If you have access in your UKBB data release, I would take a look at the effect of INHBE deficiency on body fat composition measured through DEXA - <https://biobank.ndph.ox.ac.uk/showcase/label.cgi?id=124>. This would show if INHBE is increase leg and arm adipose stores while decreasing VAT stores.
5. The splice variant rs150777893 is available in DIAMANTE, where its effect on T2D is OR 0.65 p=0.22. It may be worthwhile including this in a meta-analysis of UKBB for the T2D effect to increase power. It may similarly be worthwhile seeing if the variant is available in CARDIOGRAM for the CHD analysis and meta-analyzing with UKBB.
6. I would add "WHR" (unadjusted) and "hip circumference" to supplementary table 15 to better understand how INHBE is decrease WHRadjBMI – through increased hip circumference, given lack of association with waist circumference and BMI?
7. If you have access, I would take a look at the association of INHBE with liver fat in UKBB given the decrease in ALT observed (-0.08 SD) – this decrease is similar in magnitude to PNPLA3 I148M (+0.1 SD).
8. Line 87 – "PD3BE" is written instead of "PDE3B"

Reviewer #2 (Remarks to the Author):

Thank you for the opportunity to review this work.

The methods and conduct of the study are fairly standard for ES studies using UKBB data. My main concern relates to the novelty of this work being questionable, and it would be important that the authors clearly highlight the incremental value of their study in much more detail by referring clearly to earlier work. I am surprised that the team did not take more care to check the literature before making claims such as 'a large sequencing-based study of WHRadjBMI has not been reported to date' or 'this study provides the first evidence linking rare coding variation to abdominal adiposity' – this is simply not the case (it took me 2 minutes to find other studies) and the majority of findings of this study appear to be confirmatory rather than new (including INSR).

I would like to congratulate the authors for integrating ethnically diverse data rather than being opportunistic in the use of European-descent UKBB participants only (like so many other studies), but sadly this effort did not reveal any additional insights.

The pLOF INHBE lower WHR finding, and its liver specific expression is in my view the most interesting part of this paper. Phenotypical follow-up in human populations would ideally be expanded to rule out any potential adverse effects with greater breadth.

It would also be important that the authors clearly highlight their conflicts of interest in more detail. The group largely consists of authors with a vested commercial interest, and while this is not an issue per se in my view, it does beg the question whether this contributed to novelty being exaggerated.

Rare loss of function variants in the hepatokine gene *INHBE* protect from abdominal obesity: Response to reviewers

We thank the reviewers for their suggestions to improve the manuscript. Our response to their comments is detailed below.

Reviewer 1

1. Any thoughts on why the T2D Genes effect of *INHBE* LOF is five times as large as the UKBB effect? There appears to be significant heterogeneity between the two estimates. Different variants?

We agree that it is useful to highlight the difference in effect sizes between the two cohorts and hypothesize that differences may be due to the distinctions between each cohort (population-based for UKB vs disease-centric for T2D-GENES) or disparities in the pLOF variants present in each dataset as the reviewer suggests. Of note, a variant exclusive to T2D-GENES, Tyr253Ter, makes a large contribution to the association with WHRadjBMI and has a greater functional impact in our in vitro experiments compared to the most common variant in UKB.

We have noted the difference in effect size in the text and discussed these possible explanations (p3, lines 106-109; p4 lines 127-132; p6 lines 246-249).

2. It would be worthwhile showing the effects of the hits on WHR and BMI independently, given that conditioning on BMI will select for variants which increase or decrease BMI.

We have added these data as Supplementary Figure 6, Supplementary Table 14 and described these results in the text (p4, lines 142-147 and p5, lines 173-174).

3. I would take a look at glucose (available in UKBB) as well as HbA1c to see if *INHBE* deficiency affects glycemic traits.

We thank the reviewer for the suggestion. *INHBE* pLOF does not associate with random glucose taken at UKB assessment but does associate with fasting glucose measured in UKB primary care data. We have added these results to the manuscript (Supplementary Table 16 and p5 line 169).

4. If you have access in your UKBB data release, I would take a look at the effect of *INHBE* deficiency on body fat composition measured through DEXA. This would show if *INHBE* is increase leg and arm adipose stores while decreasing VAT stores.

We tested association of *INHBE* pLOF with body composition measured by DEXA and bioelectrical impedance as part of our deep PheWAS and did not detect any associations (p5 line 197; Supplementary Table 22). However, we are underpowered to detect associations with DEXA measurements as they are only available for a small proportion of the cohort (n= 4268 individuals in the European-ancestry population have DEXA VAT measurements including n=14 *INHBE* pLOF carriers). We estimate that to

have 80% power to detect a 0.2 SD decrease in VAT volume (an effect size similar to that of INHBE pLOF on WHRadjBMI) we would need measurements for ~122,700 individuals including 196 INHBE pLOF carriers.

5. The splice variant rs150777893 is available in DIAMANTE, where its effect on T2D is OR 0.65 p=0.22. It may be worthwhile including this in a meta-analysis of UKBB for the T2D effect to increase power. It may similarly be worthwhile seeing if the variant is available in CARDIOGRAM for the CHD analysis and meta-analyzing with UKBB.

We thank the reviewer for the suggestion but noticed that UKB data is included in the DIAMANTE analysis so meta-analysis would not be appropriate. We now note the DIAMANTE result for T2D in the text (p5, lines 179-181). Unfortunately, rs150777893 was absent from CARDIOGRAM.

6. I would add “WHR” (unadjusted) and “hip circumference” to supplementary table 15 to better understand how INHBE is decrease WHRadjBMI – through increased hip circumference, given lack of association with waist circumference and BMI?

We thank the reviewer for the suggestion and have added these data to the table (now Supplementary Table 16). We note that INHBE pLOF carriers have a non-significant trend towards decreased waist circumference (p5 lines 173-174).

7. If you have access, I would take a look at the association of INHBE with liver fat in UKBB given the decrease in ALT observed (-0.08 SD) – this decrease is similar in magnitude to PNPLA3 I148M (+0.1 SD).

We did not detect an association of INHBE pLOF with liver fat but just 18 carriers have measurements (Supplementary Table 16).

8. Line 87 – “PD3BE” is written instead of “PDE3B”

We have corrected this, thanks for pointing it out.

Reviewer 2

The methods and conduct of the study are fairly standard for ES studies using UKBB data. My main concern relates to the novelty of this work being questionable, and it would be important that the authors clearly highlight the incremental value of their study in much more detail by referring clearly to earlier work. I am surprised that the team did not take more care to check the literature before making claims such as ‘a large sequencing-based study of WHRadjBMI has not been reported to date’ or ‘this study provides the first evidence linking rare coding variation to abdominal adiposity’ – this is simply not the case (it took me 2 minutes to find other studies) and the majority of findings of this study appear to be confirmatory rather than new (including INSR).

We thank the reviewer for this feedback and have edited language in the introduction regarding the novelty of this study. We now extensively reference the study by Koprulu and colleagues which was published while this manuscript was in the submission phase as well as an exome-array study by Justice and colleagues and an early sequencing study from Kan and colleagues (p2 lines 48-54). We have removed the term “novel” when referring to the association of PLIN4 pLOF and SLC5A3 damaging missense with WHRadjBMI as these associations were previously reported (and cited previous reports) and have highlighted where others have reported associations for Mendelian disease genes, such as INSR and PLIN1, which are consistent with our findings (p2 lines 52-54 and line 67; p4 lines 136-138; added reference 41 on p8 line 322). We do feel that this study extends our knowledge of rare variation and its impact on fat distribution and, to our knowledge, is the largest sequencing-based study of WHRadjBMI reported to date. As well as the identification of INHBE as a potential therapeutic target for abdominal adiposity, novel rare coding variant associations were identified for COL5A3, ANKRD12 and KEAP1 in the sex-combined analysis and GIGYF1 and SLC35F5 in the sex-stratified analysis.

I would like to congratulate the authors for integrating ethnically diverse data rather than being opportunistic in the use of European-descent UKBB participants only (like so many other studies), but sadly this effort did not reveal any additional insights.

Thank you for this comment, we believe it is important to use data from all UKB sub-populations and not just restrict analyses to the European-ancestry population.

The pLOF INHBE lower WHR finding, and its liver specific expression is in my view the most interesting part of this paper. Phenotypical follow-up in human populations would ideally be expanded to rule out any potential adverse effects with greater breadth.

We agree with the reviewer that the association of INHBE pLOF with lower WHRadjBMI is particularly interesting and appreciate the suggestion that we look more deeply at INHBE pLOF carriers. To address the reviewer’s comment, we have added the following analyses:

1. Performed a more extensive PheWAS of INHBE. This included testing for association with an expanded set of 492 quantitative traits including NMR-derived metabolites, 1463 circulating proteins measured using Olink technology, 669 disease diagnoses, a set of 26 clinical measurements present only in UKB primary care data and body composition measurements derived from DEXA imaging and bioelectrical impedance. We now show significant and suggestive results in Table 1 and present the full PheWAS results in Figure 4 and Supplementary Tables 18-22. These results are discussed in the text on p5 (lines 185-197) and p8 (lines 297-303). Of note, was the association of INHBE pLOF with increased circulating levels of the related protein INHBC which replicated in an independent dataset.
2. Looked for evidence of excess mortality amongst INHBE pLOF carriers by performing a survival analysis in UKB (Supplementary Figure 8) and looking at the age distribution of INHBE pLOF carriers in the gnomAD database (Supplementary Table 24) (p5, lines 198-206). In these analyses we did not detect any differences for INHBE pLOF carriers compared to non-carriers.

3. Examined CNVs deleting INHBE in UKB WES and WGS data. Five carriers of CNVs disrupting INHBE were detected, and all were alive at the last UKB update with an age range of 57-81 years (Supplementary Table 25, p5 lines 206-213).

REVIEWERS' COMMENTS

Reviewer #1 (Remarks to the Author):

All of my comments have been addressed

Reviewer #2 (Remarks to the Author):

Thank you for the opportunity to comment on the response and revised version of the manuscript. I would like to thank the authors for their careful consideration of the points I raised, I have no further comments.

Response to reviewers

Reviewer #1 (Remarks to the Author):

All of my comments have been addressed

We thank the reviewer for the time they spent reviewing our manuscript.

Reviewer #2 (Remarks to the Author):

Thank you for the opportunity to comment on the response and revised version of the manuscript. I would like to thank the authors for their careful consideration of the points I raised, I have no further comments.

We thank the reviewer for the time they spent reviewing our manuscript.